# What are the Statistical Limits of Offline RL with Linear Function Approximation?

**Ruosong Wang**
Carnegie Mellon University
ruosongw@andrew.cmu.edu

**Dean P. Foster**
University of Pennsylvania and Amazon
dean@foster.net

**Sham M. Kakade**
University of Washington, Seattle and Microsoft Research
sham@cs.washington.edu

## Abstract

Offline reinforcement learning seeks to utilize offline (observational) data to guide the learning of (causal) sequential decision making strategies. The hope is that offline reinforcement learning coupled with function approximation methods (to deal with the curse of dimensionality) can provide a means to help alleviate the excessive sample complexity burden in modern sequential decision making problems. However, the extent to which this broader approach can be effective is not well understood, where the literature largely consists of sufficient conditions.

This work focuses on the basic question of what are necessary representational and distributional conditions that permit provable sample-efficient offline reinforcement learning. Perhaps surprisingly, our main result shows that even if: i) we have realizability in that the true value function of *every* policy is linear in a given set of features and 2) our off-policy data has good coverage over all features (under a strong spectral condition), any algorithm still (information-theoretically) requires a number of offline samples that is exponential in the problem horizon to nontrivially estimate the value of *any* given policy. Our results highlight that sample-efficient offline policy evaluation is not possible unless significantly stronger conditions hold; such conditions include either having low distribution shift (where the offline data distribution is close to the distribution of the policy to be evaluated) or significantly stronger representational conditions (beyond realizability).

## 1 Introduction

Offline methods (also known as off-policy methods or batch methods) are a promising methodology to alleviate the sample complexity burden in challenging reinforcement learning (RL) settings, particularly those where sample efficiency is paramount (Mandel et al., 2014; Gottesman et al., 2018; Wang et al., 2018; Yu et al., 2019). Off-policy methods are often applied together with function approximation schemes; such methods take sample transition data and reward values as inputs, and approximate the value of a target policy or the value function of the optimal policy. Indeed, many practical deep RL algorithms find their prototypes in the literature of offline RL. For example, when running on off-policy data (sometimes termed as "experience replay"), deep $Q$-networks (DQN) (Mnih et al., 2015) can be viewed as an analog of Fitted $Q$-Iteration (Gordon, 1999) with neural networks being the function approximators. More recently, there are an increasing number of both model-free (Laroche et al., 2019; Fujimoto et al., 2019; Jaques et al., 2020; Kumar et al., 2019; Agarwal et al., 2020) and model-based (Ross & Bagnell, 2012; Kidambi et al., 2020) offline RL methods, with steady improvements in performance (Fujimoto et al., 2019; Kumar et al., 2019; Wu et al., 2020; Kidambi et al., 2020).

However, despite the importance of these methods, the extent to which data reuse is possible, especially when off-policy methods are combined with function approximation, is not well understood. For example, deep $Q$-network requires millions of samples to solve certain Atari games (Mnih et al., 2015). Also important is that in some safety-critical settings, we seek guarantees when offline-

trained policies can be effective (Thomas, 2014; Thomas et al., 2019). A basic question here is that if there are fundamental statistical limits on such methods, where sample-efficient offline RL is simply not possible without further restrictions on the problem.

In the context of supervised learning, it is well-known that empirical risk minimization is sample-efficient if the hypothesis class has bounded complexity. For example, suppose the agent is given a $d$-dimensional feature extractor, and the ground truth labeling function is a (realizable) linear function with respect to the feature mapping. Here, it is well-known that a polynomial number of samples in $d$ suffice for a given target accuracy. Furthermore, in this realizable case, provided the training data has a good feature coverage, then we will have good accuracy against any test distribution.[1]

In the more challenging offline RL setting, it is unclear if sample-efficient methods are possible, even under analogous assumptions. This is our motivation to consider the following question:

**What are the statistical limits for offline RL with linear function approximation?**

Here, one may hope that value estimation for a given policy is possible in the offline RL setting under the analogous set of assumptions that enable sample-efficient supervised learning, i.e., 1) (realizability) the features can perfectly represent the value functions and 2) (good coverage) the feature covariance matrix of our off-policy data has lower bounded eigenvalues.

The extant body of provable methods on offline RL either make representational assumptions that are far stronger than realizability or assume distribution shift conditions that are far stronger than having coverage with regards to the spectrum of the feature covariance matrix of the data distribution. For example, Szepesvári & Munos (2005) analyze offline RL methods by assuming a representational condition where the features satisfy (approximate) closedness under Bellman updates, which is a far stronger representation condition than realizability. Recently, Xie & Jiang (2020a) propose a offline RL algorithm that only requires realizability as the representation condition. However, the algorithm in (Xie & Jiang, 2020a) requires a more stringent data distribution condition. Whether it is possible to design a sample-efficient offline RL method under the realizability assumption and a reasonable data coverage assumption — an open problem in (Chen & Jiang, 2019) — is the focus of this work.

**Our Contributions.** Perhaps surprisingly, our main result shows that, under only the above two assumptions, it is information-theoretically not possible to design a sample-efficient algorithm to non-trivially estimate the value of a given policy. The following theorem is an informal version of the result in Section 4.

**Theorem 1.1** (Informal). *In the offline RL setting, suppose the data distributions have (polynomially) lower bounded eigenvalues, and the Q-functions of* every *policy are linear with respect to a given feature mapping. Any algorithm requires an exponential number of samples in the horizon $H$ to output a non-trivially accurate estimate of the value of* any *given policy $\pi$, with constant probability.*

This hardness result states that even if the $Q$-functions of *all* polices are linear with respect to the given feature mapping, we still require an exponential number of samples to evaluate *any* given policy. Note that this representation condition is significantly stronger than assuming realizability with regards to only a single target policy; it assumes realizability for all policies. Regardless, even under this stronger representation condition, it is hard to evaluate any policy, as specified in our hardness result.

This result also formalizes a key issue in offline reinforcement learning with function approximation: geometric error amplification. To better illustrate the issue, in Section 5, we analyze the classical Least-Squares Policy Evaluation (LSPE) algorithm under the realizability assumption, which demonstrates how the error propagates as the algorithm proceeds. Here, our analysis shows that, if we only rely on the realizability assumption, then a far more stringent condition is required for sample-efficient offline policy evaluation: the off-policy data distribution must be quite close to the distribution induced by the policy to be evaluated.

---

[1]Specifically, if the features have a uniformly bounded norm and if the minimum eigenvalue of the feature covariance matrix of our data is bounded away from 0, say by $1/\text{poly}(d)$, then we have good accuracy on any test distribution. See Assumption 2 and the comments thereafter.

Our results highlight that sample-efficient offline RL is simply not possible unless either the distribution shift condition is sufficiently mild or we have stronger representation conditions that go well beyond realizability. See Section 5 for more details.

Furthermore, our hardness result implies an exponential separation on the sample complexity between offline RL and supervised learning, since supervised learning (which is equivalent to offline RL with $H = 1$) is possible with polynomial number of samples under the same set of assumptions.

A few additional points are worth emphasizing with regards to our lower bound construction:

- Our results imply that Least-Squares Policy Evaluation (LSPE, i.e., using Bellman backups with linear regression) will fail. Interestingly, while LSPE will provide an unbiased estimator, our results imply that it will have exponential variance in the problem horizon.
- Our construction is simple and does not rely on having a large state or action space: the size of the state space is only $O(d \cdot H)$ where $d$ is the feature dimension and $H$ is the planning horizon, and the size of the action space is only is 2. This stands in contrast to other RL lower bounds, which typically require state spaces that are exponential in the problem horizon (e.g. see (Du et al., 2020)).
- We provide two hard instances, one with a sparse reward (and stochastic transitions) and another with deterministic dynamics (and stochastic rewards). These two hard instances jointly imply that both the estimation error on reward values and the estimation error on the transition probabilities could be geometrically amplified in offline RL.
- Of possibly broader interest is that our hard instances are, to our knowledge, the first concrete examples showing that geometric error amplification is real in RL problems (even with realizability). While this is a known concern in the analysis of RL algorithms, there have been no concrete examples exhibiting such behavior under only a realizability assumption.

## 2 RELATED WORK

We now survey prior work on offline RL, largely focusing on theoretical results. We also discuss results on the error amplification issue in RL. Concurrent to this work, Xie & Jiang (2020a) propose a offline RL algorithm under the realizability assumption, which requires stronger distribution shift conditions. We will discuss this work shortly.

**Existing Algorithms and Analysis.** Offline RL with value function approximation is closely related to Approximate Dynamic Programming (Bertsekas & Tsitsiklis, 1995). Existing work (Munos, 2003; Szepesvári & Munos, 2005; Antos et al., 2008; Munos & Szepesvári, 2008; Tosatto et al., 2017; Xie & Jiang, 2020b; Duan & Wang, 2020) that analyze the sample complexity of approximate dynamic programming-based approaches usually make the following two categories of assumptions: (i) representation conditions that assume the function class approximates the value functions well and (ii) distribution shift conditions that assume the given data distribution has sufficient coverage over the state-action space. As mentioned in the introduction, the desired representation condition would be realizability, which only assumes the value function of the policy to be evaluated lies in the function class (for the case of offline policy evaluation) or the optimal value function lies in the function class (for the case of finding near-optimal policies), and existing works usually make stronger assumptions. For example, Szepesvári & Munos (2005); Duan & Wang (2020) assume (approximate) closedness under Bellman updates, which is much stronger than realizability. Whether it is possible to design a sample-efficient offline RL method under the realizability assumption and reasonable data coverage assumption, is left as an open problem in (Chen & Jiang, 2019).

To measure the coverage over the state-action space of the given data distribution, existing works assume the concentrability coefficient (introduced by Munos (2003)) to be bounded. The concentrability coefficient, informally speaking, is the largest possible ratio between the probability for a state-action pair $(s, a)$ to be visited by a policy, and the probability that $(s, a)$ appears on the data distribution. Since we work with linear function approximation in this work, we measure the distribution shift in terms of the spectrum of the feature covariance matrices (see Assumption 2), which is a well-known sufficient condition in the context of supervised learning and is much more natural for the case of linear function approximation.

Concurrent to this work, Xie & Jiang (2020a) propose an algorithm that works under the realizability assumption instead of other stronger representation conditions used in prior work. However, the algorithm in (Xie & Jiang, 2020a) requires a much stronger data distribution condition which assumes a stringent version of concentrability coefficient introduced by (Munos, 2003) to be bounded. In contrast, in this work we measure the distribution shift in terms of the spectrum of the feature covariance matrix of the data distribution, which is more natural than the concentrability coefficient for the case of linear function approximation.

Recently, there has been great interest in approaching offline policy evaluation (Precup, 2000) via importance sampling. For recent work on this topic, see (Dudík et al., 2011; Mandel et al., 2014; Thomas et al., 2015; Li et al., 2015; Jiang & Li, 2016; Thomas & Brunskill, 2016; Guo et al., 2017; Wang et al., 2017; Liu et al., 2018; Farajtabar et al., 2018; Xie et al., 2019; Kallus & Uehara, 2019; Liu et al., 2019; Uehara & Jiang, 2019; Kallus & Uehara, 2020; Jiang & Huang, 2020; Feng et al., 2020). Offline policy evaluation with importance sampling incurs exponential variance in the planning horizon when the behavior policy is significantly different from the policy to be evaluated. Bypassing such exponential dependency requires non-trivial function approximation assumptions (Jiang & Huang, 2020; Feng et al., 2020; Liu et al., 2018). Finally, Kidambi et al. (2020) provide a model-based offline RL algorithm, with a theoretical analysis based on hitting times.

**Hardness Results.** Algorithm-specific hardness results have been known for a long time in the literature of Approximate Dynamic Programming. See Chapter 4 in (Van Roy, 1994) and also (Gordon, 1995; Tsitsiklis & Van Roy, 1996). These works demonstrate that certain approximate dynamic programming-based methods will diverge on hard cases. However, such hardness results only hold for a restricted class of algorithms, and to demonstrate the fundamental difficulty of offline RL, it is more desirable to obtain information-theoretic lower bounds as initiated by Chen & Jiang (2019).

Existing (information-theoretic) exponential lower bounds (Krishnamurthy et al., 2016; Sun et al., 2017; Chen & Jiang, 2019) usually construct unstructured MDPs with an exponentially large state space. Du et al. (2020) prove an exponential lower bound under the assumption that the optimal $Q$-function is approximately linear. The condition that the optimal $Q$-function is only *approximately* linear is crucial for the hardness result in Du et al. (2020). The techniques in (Du et al., 2020) are later generalized to other settings (Kumar et al., 2020; Wang et al., 2020; Mou et al., 2020).

**Error Amplification In RL.** Error amplification induced by distribution shift and long planning horizon is a known issue in the theoretical analysis of RL algorithms. See (Gordon, 1995; 1996; Munos & Moore, 1999; Ormoneit & Sen, 2002; Kakade, 2003; Zanette et al., 2019) for papers on this topic and additional assumptions that mitigate this issue. Error amplification in offline RL is also observed in empirical works (see e.g. (Fujimoto et al., 2019)). In this work, we provide the first information-theoretic lower bound showing that geometric error amplification is real in offline RL.

## 3    THE OFFLINE POLICY EVALUATION PROBLEM

Throughout this paper, for a given integer $H$, we use $[H]$ to denote the set $\{1, 2, \ldots, H\}$.

**Episodic Reinforcement Learning.** Let $M = (\mathcal{S}, \mathcal{A}, P, R, H)$ be a *Markov Decision Process* (MDP) where $\mathcal{S}$ is the state space, $\mathcal{A}$ is the action space, $P : \mathcal{S} \times \mathcal{A} \to \Delta(\mathcal{S})$ is the transition operator which takes a state-action pair and returns a distribution over states, $R : \mathcal{S} \times \mathcal{A} \to \Delta(\mathbb{R})$ is the reward distribution, $H \in \mathbb{Z}_+$ is the planning horizon. For simplicity, we assume a fixed initial state $s_1 \in \mathcal{S}$. A (stochastic) policy $\pi : \mathcal{S} \to \Delta(\mathcal{A})$ chooses an action $a$ randomly based on the current state $s$. The policy $\pi$ induces a (random) trajectory $s_1, a_1, r_1, s_2, a_2, r_2, \ldots, s_H, a_H, r_H$, where $a_1 \sim \pi_1(s_1), r_1 \sim R(s_1, a_1), s_2 \sim P(s_1, a_1), a_2 \sim \pi_2(s_2)$, etc. To streamline our analysis, for each $h \in [H]$, we use $\mathcal{S}_h \subseteq \mathcal{S}$ to denote the set of states at level $h$, and we assume $\mathcal{S}_h$ do not intersect with each other. We assume, almost surely, that $r_h \in [-1, 1]$ for all $h \in [H]$.

**Value Functions.** Given a policy $\pi$, $h \in [H]$ and $(s, a) \in \mathcal{S}_h \times \mathcal{A}$, define $Q_h^\pi(s, a) = \mathbb{E}\left[\sum_{h'=h}^H r_{h'} \mid s_h = s, a_h = a, \pi\right]$ and $V_h^\pi(s) = \mathbb{E}\left[\sum_{h'=h}^H r_{h'} \mid s_h = s, \pi\right]$. For a policy $\pi$, we define $V^\pi = V_1^\pi(s_1)$ to be the value of $\pi$ from the fixed initial state $s_1$.

**Linear Function Approximation.** When applying linear function approximation schemes, it is commonly assumed that the agent is given a feature extractor $\phi : \mathcal{S} \times \mathcal{A} \to \mathbb{R}^d$ which can either be hand-crafted or a pre-trained neural network that transforms a state-action pair to a $d$-dimensional embedding, and the $Q$-functions can be predicted by linear functions of the features. In this paper, we are interested in the following *realizability* assumption.

**Assumption 1** (Realizable Linear Function Approximation). *For every policy* $\pi : \mathcal{S} \to \Delta(\mathcal{A})$*, there exists* $\theta_1^\pi, \ldots \theta_H^\pi \in \mathbb{R}^d$ *such that for all* $(s, a) \in \mathcal{S} \times \mathcal{A}$ *and* $h \in [H]$*,* $Q_h^\pi(s, a) = (\theta_h^\pi)^\top \phi(s, a)$*.*

Note that our assumption is much stronger than assuming realizability with regards to a single policy $\pi$ (say the policy that we wish to evaluate); our assumption imposes realizability for all policies.

**Offline Reinforcement Learning.** This paper is concerned with the offline RL setting. In this setting, the agent does not have direct access to the MDP and instead is given access to data distributions $\{\mu_h\}_{h=1}^H$ where for each $h \in [H]$, $\mu_h \in \Delta(\mathcal{S}_h \times \mathcal{A})$. The inputs of the agent are $H$ datasets $\{D_h\}_{h=1}^H$, and for each $h \in [H]$, $D_h$ consists i.i.d. samples of the form $(s, a, r, s') \in \mathcal{S}_h \times \mathcal{A} \times \mathbb{R} \times \mathcal{S}_{h+1}$ tuples, where $(s, a) \sim \mu_h$, $r \sim r(s, a)$, $s' \sim P(s, a)$.

In this paper, we focus on the *offline policy evaluation* problem with linear function approximation: given a policy $\pi : \mathcal{S} \to \Delta(\mathcal{A})$ and a feature extractor $\phi : \mathcal{S} \times \mathcal{A} \to \mathbb{R}^d$, the goal is to output an accurate estimate of the value of $\pi$ (i.e., $V^\pi$) approximately, using the collected datasets $\{D_h\}_{h=1}^H$, with as few samples as possible.

**Notation.** For a vector $x \in \mathbb{R}^d$, we use $\|x\|_2$ to denote its $\ell_2$ norm. For a positive semidefinite matrix $A$, we use $\|A\|_2$ to denote its operator norm, and $\sigma_{\min}(A)$ to denote its smallest eigenvalue. For two positive semidefinite matrices $A$ and $B$, we write $A \succeq B$ to denote the Löwner partial ordering of matrices, i.e, $A \succeq B$ if and only if $A - B$ is positive semidefinite. For a policy $\pi : \mathcal{S} \to \Delta(\mathcal{A})$, we use $\mu_h^\pi$ to denote the marginal distribution of $s_h$ under $\pi$, i.e., $\mu_h^\pi(s) = \Pr[s_h = s \mid \pi]$. For a vector $x \in \mathbb{R}^d$ and a positive semidefinite matrix $A \in \mathbb{R}^{d \times d}$, we use $\|x\|_A$ to denote $\sqrt{x^\top A x}$.

## 4 THE LOWER BOUND: REALIZABILITY AND COVERAGE ARE INSUFFICIENT

We now present our main hardness result for offline policy evaluation with linear function approximation. It should be evident that without feature coverage in our dataset, realizability alone is clearly not sufficient for sample-efficient estimation. Here, we will make the strongest possible assumption, with regards to the conditioning of the feature covariance matrix.

**Assumption 2** (Feature Coverage). *For all* $(s, a) \in \mathcal{S} \times \mathcal{A}$*, assume our feature map is bounded such that* $\|\phi(s, a)\|_2 \leq 1$*. Furthermore, suppose for each* $h \in [H]$*, the data distributions* $\mu_h$ *satisfy the following minimum eigenvalue condition:* $\sigma_{\min}\left(\mathbb{E}_{(s,a)\sim\mu_h}[\phi(s, a)\phi(s, a)^\top]\right) = 1/d$.[2]

Clearly, for the case where $H = 1$, the realizability assumption (Assumption 1), and feature coverage assumption (Assumption 2) imply that the ordinary least squares estimator will accurately estimate $\theta_1$.[3] Our main result now shows that these assumptions are not sufficient for offline policy evaluation for long horizon problems.

**Theorem 4.1.** *Suppose Assumption 2 holds. Fix an algorithm that takes as input both a policy and a feature mapping. There exists a (deterministic) MDP satisfying Assumption 1, such that for* any *policy* $\pi : \mathcal{S} \to \Delta(\mathcal{A})$*, the algorithm requires* $\Omega((d/2)^H)$ *samples to output the value of* $\pi$ *up to constant additive approximation error with probability at least* 0.9.

Although we focus on offline policy evaluation in this work, our hardness result also holds for finding near-optimal policies under Assumption 1 in the offline RL setting. Below we give a simple reduction. At the initial state, if the agent chooses action $a_1$, then the agent receives a fixed reward value (say 0.5) and terminates. If the agent chooses action $a_2$, then the agent transits to our hard

---

[2]Note that $1/d$ is the largest possible minimum eigenvalue due to that, for any data distribution $\widetilde{\mu}_h$, $\sigma_{\min}(\mathbb{E}_{(s,a)\sim\widetilde{\mu}_h}[\phi(s, a)\phi(s, a)^\top]) \leq 1/d$ since $\|\phi(s, a)\|_2 \leq 1$ for all $(s, a) \in \mathcal{S} \times \mathcal{A}$.

[3]For $H = 1$, the ordinary least squares estimator will satisfy that $\|\theta_1 - \widehat{\theta}_{\text{OLS}}\|_2^2 \leq O(d/n)$ with high probability. See e.g. (Hsu et al., 2012b).

instance. In order to find a policy with suboptimality at most $0.5$, the agent must evaluate the value of the optimal policy in our hard instance up to an error of $0.5$, and hence the hardness result holds.

**Remark 1** (The sparse reward case). As stated, the theorem uses a deterministic MDP (with stochastic rewards). See Appendix C for another hard case where the transition is stochastic and the reward is deterministic and sparse (only occurring at two states at $h = H$).

**Remark 2** (Least-Squares Policy Evaluation (LSPE) has exponential variance). For offline policy evaluation with linear function approximation, the most naïve algorithm here would be LSPE, i.e., using ordinary least squares (OLS) to estimate $\theta^\pi$, starting at level $h = H$ and then proceeding backwards to level $h = 1$, using the plug-in estimator from the previous level. Here, LSPE will provide an unbiased estimate (provided the feature covariance matrices are full rank, which will occur with high probability). As a direct corollary, the above theorem implies that LSPE has exponential variance in $H$. See Section 5 for a more detailed discussion on LSPE. More generally, our theorem implies that there is no estimator that can avoid such exponential dependence in the offline setting.

**Remark 3** (Least-Squares Value Iteration (LSVI) versus Least-Squares Policy Iteration (LSPI)). In the offline setting, under Assumptions 1 and 2, in order to find a near-optimal policy, the most naïve algorithm would be LSVI, i.e., using ordinary least squares (OLS) to estimate $\theta^*$, starting at level $h = H$ and then proceeding backwards to level $h = 1$, using the plug-in estimator from the previous level and the bellman operator. The above theorem implies that LSVI will require an exponential number of samples to find a near-optimal policy. On the other hand, if the regression targets are collected by using rollouts (i.e. on-policy sampling) as in LSPI (Lagoudakis & Parr, 2003), then a polynomial number of samples suffice. See Section D in (Du et al., 2020) for an analysis. Thus, Theorem 4.1 implies an exponential separation on the sample complexity between LSVI and LSPI. Of course, LSPI requires adaptive data samples and thus does not work in the offline setting.

One may wonder if Theorem 4.1 still holds when the data distributions $\{\mu_h\}_{h=1}^H$ are induced by a policy. In Appendix C, we prove another exponential sample complexity lower bound under the additional assumption that the data distributions are induced by a fixed policy $\pi$. However, under such an assumption, it is impossible to prove a hardness result as strong as Theorem 4.1 (which shows that evaluating *any* policy is hard), since one can at least evaluate the policy $\pi$ that induces the data distributions. Nevertheless, we are able to prove the hardness of offline policy evaluation, under a weaker version of Assumption 1. See Appendix C for more details.

In the remaining part of this section, we give the hard instance construction and the proof of Theorem 4.1. We use $d$ the denote the feature dimension, and we assume $d$ is even for simplicity. We use $\hat{d}$ to denote $d/2$ for convenience. We also provide an illustration of the construction in Figure 1.

**State Space, Action Space and Transition Operator.** The action space $\mathcal{A} = \{a_1, a_2\}$. For each $h \in [H]$, $\mathcal{S}_h$ contains $\hat{d} + 1$ states $s_h^1, s_h^2, \ldots, s_h^{\hat{d}}$ and $s_h^{\hat{d}+1}$. For each $h \in [H-1]$, for each $c \in \{1, 2, \ldots, \hat{d}+1\}$, we have $P(s_h^c, a_1) = s_{h+1}^{\hat{d}+1}$ and $P(s_h^c, a_1) = s_{h+1}^c$.

**Reward Distributions.** Let $0 \le r_0 \le \hat{d}^{-H/2}$ be a parameter to be determined. For each $(h, c) \in [H-1] \times [\hat{d}]$ and $a \in \mathcal{A}$, we set $R(s_h^c, a) = 0$ and $R(s_h^{\hat{d}+1}, a) = r_0 \cdot (\hat{d}^{1/2} - 1) \cdot \hat{d}^{(H-h)/2}$. For the last level, for each $c \in [\hat{d}]$ and $a \in \mathcal{A}$, we set $R(s_H^c, a) = \begin{cases} 1 & \text{with probability } (1+r_0)/2 \\ -1 & \text{with probability } (1-r_0)/2 \end{cases}$ so that $\mathbb{E}[R(s_H^c, a)] = r_0$. Moreover, for all actions $a \in \mathcal{A}$, $R(s_H^{\hat{d}+1}, a) = r_0 \cdot \hat{d}^{1/2}$.

**Feature Mapping.** Let $e_1, e_2, \ldots, e_d$ be a set of orthonormal vectors in $\mathbb{R}^d$. Here, one possible choice is to set $e_1, e_2, \ldots, e_d$ to be the standard basis vectors. For each $(h, c) \in [H] \times [\hat{d}]$, we set $\phi(s_h^c, a_1) = e_c, \phi(s_h^c, a_2) = e_{c+\hat{d}}$, and $\phi(s_h^{\hat{d}+1}, a) = \sum_{c \in \hat{d}} e_c/\hat{d}^{1/2}$ for all $a \in \mathcal{A}$.

**Verifying Assumption 1.** The following lemma shows that Assumption 1 holds for our construction. The formal proof can be found in Appendix A.

**Lemma 4.2.** *For every policy $\pi : \mathcal{S} \to \Delta(\mathcal{A})$, for each $h \in [H]$, for all $(s, a) \in \mathcal{S}_h \times \mathcal{A}$, we have $Q_h^\pi(s, a) = (\theta_h^\pi)^\top \phi(s, a)$ for some $\theta_h^\pi \in \mathbb{R}^d$.*

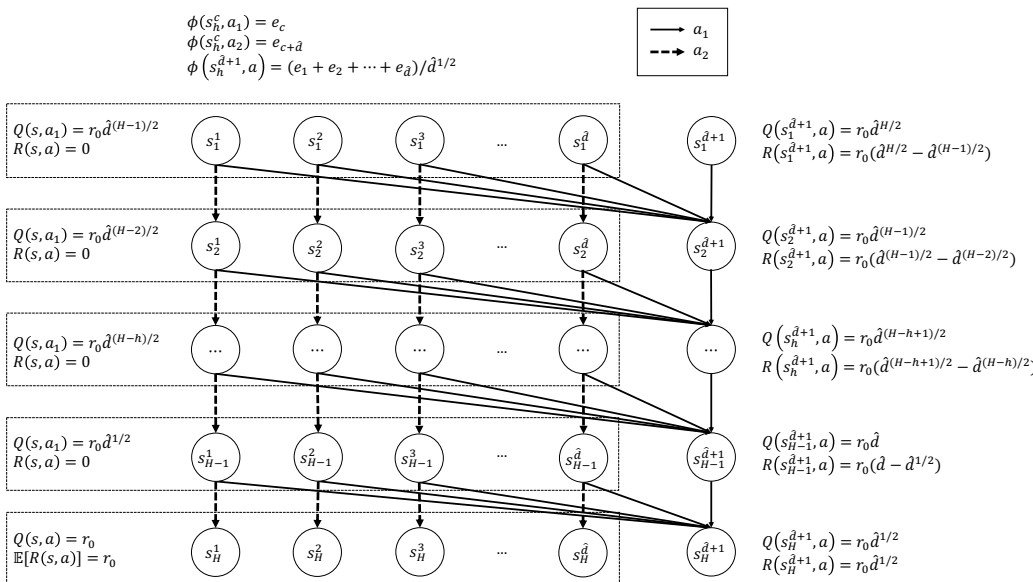

Figure 1: An illustration of the hard instance. Recall that $\hat{d} = d/2$. States on the top are those in the first level ($h = 1$), while states at the bottom are those in the last level ($h = H$). Solid line (with arrow) corresponds to transitions associated with action $a_1$, while dotted line (with arrow) corresponds to transitions associated with action $a_2$. For each level $h \in [H]$, reward values and $Q$-values associated with $s_h^1, s_h^2, \ldots, s_h^{\hat{d}}$ are marked on the left, while reward values and $Q$-values associated with $s_h^{\hat{d}+1}$ are mark on the right. Rewards and transitions are all deterministic, except for the reward distributions associated with $s_H^1, s_H^2, \ldots, s_H^{\hat{d}}$. We mark the expectation of the reward value when it is stochastic. For each level $h \in [H]$, for the data distribution $\mu_h$, the state is chosen uniformly at random from those states in the dashed rectangle, i.e., $\{s_h^1, s_h^2, \ldots, s_h^{\hat{d}}\}$, while the action is chosen uniformly at random from $\{a_1, a_2\}$. Suppose the initial state is $s_1^{\hat{d}+1}$. When $r_0 = 0$, the value of the policy is 0. When $r_0 = \hat{d}^{-H/2}$, the value of the policy is $r_0 \cdot \hat{d}^{H/2} = 1$.

**The Data Distributions.** For each level $h \in [H]$, the data distribution $\mu_h$ is a uniform distribution over $\{(s_h^1, a_1), (s_h^1, a_2), (s_h^2, a_1), (s_h^2, a_2), \ldots, (s_h^{\hat{d}}, a_1), (s_h^{\hat{d}}, a_2)\}$. Notice that $(s_h^{\hat{d}+1}, a)$ is *not* in the support of $\mu_h$ for all $a \in \mathcal{A}$. It can be seen that, $\mathbb{E}_{(s,a) \sim \mu_h} [\phi(s,a)\phi(s,a)^\top] = \frac{1}{d} \sum_{c=1}^{d} e_c e_c^\top = \frac{1}{d} I$.

**The Lower Bound.** We show that it is information-theoretically hard for any algorithm to distinguish the case $r_0 = 0$ and $r_0 = \hat{d}^{-H/2}$. We fix the initial state to be $s_1^{\hat{d}+1}$, and consider any policy $\pi$. When $r_0 = 0$, all reward values will be zero, and thus the value of $\pi$ is zero. On the other hand, when $r_0 = \hat{d}^{-H/2}$, the value of $\pi$ would be $r_0 \cdot \hat{d}^{H/2} = 1$. Thus, if the algorithm approximates the value of the policy up to an error of $1/2$, then it must distinguish the case that $r_0 = 0$ and $r_0 = \hat{d}^{-H/2}$.

We first notice that for the case $r_0 = 0$ and $r_0 = \hat{d}^{-H/2}$, the data distributions $\{\mu_h\}_{h=1}^H$, the feature mapping $\phi : \mathcal{S} \times \mathcal{A} \to \mathbb{R}^d$, the policy $\pi$ to be evaluated and the transition operator $P$ are the same. Thus, in order to distinguish the case $r_0 = 0$ and $r_0 = \hat{d}^{-H/2}$, the only way is to query the reward distribution by using sampling taken from the data distributions. For all state-action pairs $(s,a)$ in the support of the data distributions of the first $H - 1$ levels, the reward distributions will be identical. This is because for all $s \in \mathcal{S}_h \setminus \{s_h^{\hat{d}+1}\}$ and $a \in \mathcal{A}$, we have $R(s,a) = 0$. For the case $r_0 = 0$ and $r_0 = \hat{d}^{-H/2}$, for all state-action pairs $(s,a)$ in the support of the data distribution of the last level, $R(s,a) = \begin{cases} 1 & \text{with probability } (1 + r_0)/2 \\ -1 & \text{with probability } (1 - r_0)/2 \end{cases}$. Therefore, to distinguish

---

**Algorithm 1** Least-Squares Policy Evaluation

---

1: **Input:** policy $\pi$ to be evaluated, number of samples $N$, regularization parameter $\lambda > 0$
2: Let $Q_{H+1}(\cdot, \cdot) = 0$ and $V_{H+1}(\cdot) = 0$
3: **for** $h = H, H-1, \ldots, 1$ **do**
4:     Take samples $(s_h^i, a_h^i) \sim \mu_h$, $r_h^i \sim r(s_h^i, a_h^i)$ and $\overline{s}_h^i \sim P(s_h^i, a_h^i)$ for each $i \in [N]$
5:     Let $\hat{\Lambda}_h = \sum_{i \in [N]} \phi(s_h^i, a_h^i)\phi(s_h^i, a_h^i)^\top + \lambda I$
6:     Let $\hat{\theta}^h = \hat{\Lambda}_h^{-1}\left(\sum_{i=1}^N \phi(s_h^i, a_h^i) \cdot (r_h^i + \hat{V}_{h+1}(\overline{s}_h^i))\right)$
7:     Let $\hat{Q}_h(\cdot, \cdot) = \phi(\cdot, \cdot)^\top \hat{\theta}_h$ and $\hat{V}_h(\cdot) = \hat{Q}(\cdot, \pi(\cdot))$

---

the case that $r_0 = 0$ and $r_0 = \hat{d}^{-H/2}$, the agent needs to distinguish two reward distributions
$r_1 = \begin{cases} 1 & \text{with probability } 1/2 \\ -1 & \text{with probability } 1/2 \end{cases}$ and $r_2 = \begin{cases} 1 & \text{with probability } (1 + \hat{d}^{-H/2})/2 \\ -1 & \text{with probability } (1 - \hat{d}^{-H/2})/2 \end{cases}$ . Now we invoke Lemma B.1 in Section B by setting $\varepsilon = \hat{d}^{-H/2}/2$ and $\delta = 0.9$. By Lemma B.1, in order to distinguish $r_1$ and $r_2$ with probability at least 0.9, any algorithm requires $\Omega(\hat{d}^H)$ samples.

**Remark 4.** The key in our construction is the state $s_h^{\hat{d}+1}$ in each level, whose feature vector is defined to be $\sum_{c \in \hat{d}} e_c / \hat{d}^{1/2}$. In each level, $s_h^{\hat{d}+1}$ amplifies the $Q$-value by a $\hat{d}^{1/2}$ factor, due to the linearity of the $Q$-function. After all the $H$ levels, the value will be amplified by a $\hat{d}^{H/2}$ factor. Since $s_h^{\hat{d}+1}$ is not in the support of the data distribution, the only way to estimate the value of the policy is to estimate the expected reward value in the last level. Our construction forces the estimation error of the last level to be amplified exponentially and thus implies an exponential lower bound.

## 5   Upper Bounds: Low Distribution Shift or Policy Completeness are Sufficient

In order to illustrate the error amplification issue and discuss conditions that permit sample-efficient offline RL, in this section, we analyze Least-Squares Policy Evaluation when applied to the offline policy evaluation problem under the realizability assumption. The algorithm is presented in Algorithm 1. For simplicity here we assume the policy $\pi$ to be evaluated is deterministic.

**Notation.** For each $h \in [H]$, define $\Lambda_h = \mathbb{E}_{(s,a) \sim \mu_h}\left[\phi(s,a)\phi(s,a)^\top\right]$ to be the feature covariance matrix at level $h$. For each $h \in [H-1]$, define $\overline{\Lambda}_{h+1} = \mathbb{E}_{(s,a) \sim \mu_h, \overline{s} \sim P(\cdot|s,a)}\left[\phi(\overline{s}, \pi(\overline{s}))\phi(\overline{s}, \pi(\overline{s}))^\top\right]$ to be the feature covariance matrix of the one-step lookahead distribution at level $h$. Moreover, define $\overline{\Lambda}_1 = \phi(s_1, \pi(s_1))\phi(s_1, \pi(s_1))^\top$. We define $\Phi_h$ to be a $N \times d$ matrix, whose $i$-th row is $\phi(s_h^i, a_h^i)$, and define $\overline{\Phi}_{h+1}$ to be another $N \times d$ matrix whose $i$-th row is $\phi(\overline{s}_h^i, \pi(\overline{s}_h^i))$. For each $h \in [H]$ and $i \in [N]$, define $\xi_h^i = r_h^i + V(\overline{s}_h^i) - Q(s_h^i, a_h^i)$. We use $\xi_h$ to denote a vector whose $i$-th entry is $\xi_h^i$.

Now we present a general lemma that characterizes the estimation error of Algorithm 1 by an *equality*. The proof can be found in Appendix D. Later, we apply this general lemma to special cases.

**Lemma 5.1.** *Suppose $\lambda > 0$ in Algorithm 1, and for the given policy $\pi$, there exists $\theta_1, \theta_2, \ldots, \theta_d \in \mathbb{R}^d$ such that for each $h \in [H]$, $Q_h^\pi(s,a) = \phi(s,a)^\top \theta_h$ for all $(s,a) \in \mathcal{S}_h \times \mathcal{A}$. Then we have*

$$(Q^\pi(s_1, \pi(s_1)) - \hat{Q}(s_1, \pi(s_1)))^2 = \left\| \sum_{h=1}^H \hat{\Lambda}_1^{-1}\Phi_1^\top \overline{\Phi}_2 \hat{\Lambda}_2^{-1}\Phi_2^\top \cdots (\hat{\Lambda}_h^{-1}\Phi_h^\top \xi_h - \lambda \hat{\Lambda}_h^{-1}\theta_h) \right\|_{\overline{\Lambda}_1}^2 . \quad (1)$$

Now we consider two special cases where the estimation error in Equation (1) can be upper bounded.

**Low Distribution Shift.** The first special we focus on is the case where the distribution shift between the data distributions and the distribution induced by the policy to be evaluated is low. To measure the distribution shift formally, our main assumption is as follows.

**Assumption 3.** *We assume that for each $h \in [H]$, there exists $C_h \geq 1$ such that $\overline{\Lambda}_h \preceq C_h \Lambda_h$.*

**Remark 5.** For each $h \in [H]$, if $\sigma_{\min}(\Lambda_h) \succeq \frac{1}{C_h} I$ for some $C_h \geq 1$, we have $\overline{\Lambda}_h \preceq I \preceq C_h \Lambda_h$. Therefore, Assumption 3 can be replaced with the assumption that $C_h \Lambda_h \succeq I$. However, we stick to the original version of Assumption 3 as it gives a tighter characterization of the distribution shift.

Now we state the theoretical guarantee of Algorithm 1. The proof can be found in Appendix D.

**Theorem 5.2.** *Suppose for the given policy $\pi$, there exists $\theta_1, \theta_2, \ldots, \theta_d \in \mathbb{R}^d$ such that for each $h \in [H]$, $Q_h^\pi(s, a) = \phi(s, a)^\top \theta_h$ for all $(s, a) \in \mathcal{S}_h \times \mathcal{A}$ and $\|\theta_h\|_2 \leq H\sqrt{d}$.[4] Let $\lambda = CH\sqrt{d \log(dH/\delta)N}$ for some $C > 0$. With probability at least $1 - \delta$, for some $c > 0$, $(Q_1^\pi(s_1, \pi(s_1)) - \hat{Q}_1(s_1, \pi(s_1)))^2 \leq c \cdot \prod_{h=1}^H C_h \cdot dH^5 \cdot \sqrt{d \log(dH/\delta)/N}$.*

**Remark 6.** The factor $\prod_{h=1}^H C_h$ in Theorem 5.2 implies that the estimation error will be amplified *geometrically*. Now we discuss how the error is amplified when running Algorithm 1 on the instance in Section 4 to better illustrate the issue. If we run Algorithm 1 on the hard instance in Section 4, when $h = H$, the estimation error on $V(s_H^c)$ would be roughly $N^{-1/2}$ for each $c \in [\hat{d}]$. When using the linear predictor at level $H$ to predict the value of $s_H^*$, the error will be amplified by $\hat{d}^{1/2}$. When $h = H - 1$, the dataset contains only $s_{H-1}^c$ for $c \in [\hat{d}]$, and the estimation error on the value of $s_{H-1}^c$ will be the same as that of $s_H^*$, which is roughly $(\hat{d}/N)^{1/2}$. Again, the estimation error on the value of $s_{H-1}^*$ will be $(\hat{d}^2/N)^{1/2}$ when using the linear predictor at level $H - 1$. The error will eventually be amplified by a factor of $\hat{d}^{H/2}$, which corresponds to the factor $\prod_{h=1}^H C_h$ in Theorem 5.2.

**Policy Completeness.** In offline RL, another representation condition is closedness under Bellman update (Szepesvári & Munos, 2005; Duan & Wang, 2020), which is stronger than realizability. In the context of offline policy evaluation, we have the following policy completeness assumption.

**Assumption 4.** *For the given policy $\pi$, for any $h > 1$ and $\theta_h \in \mathbb{R}^d$, there exists $\theta' \in \mathbb{R}^d$ such that for any $(s, a) \in \mathcal{S}_{h-1} \times \mathcal{A}$, $\mathbb{E}[R(s, a)] + \sum_{s' \in \mathcal{S}_h} P(s' \mid s, a) \phi(s', \pi(s'))^\top \theta_h = \phi(s, a)^\top \theta'$.*

Under Assumption 4 and the assumption that $\sigma_{\min}(\Lambda_h) \geq \lambda_0$ for all $h \in [H]$ for some $\lambda_0 > 0$, Duan & Wang (2020) have shown that for Algorithm 1, by taking $N = \text{poly}(H, d, 1/\varepsilon, 1/\lambda_0)$, we have $(Q_1^\pi(s_1, \pi(s_1)) - \hat{Q}_1(s_1, \pi(s_1)))^2 \leq \varepsilon$. We refer interested readers to (Duan & Wang, 2020).

We remark that the above analysis again implies that geometric error amplification is a real issue in offline RL, and sample-efficient offline RL is impossible unless the distribution shift is sufficiently low, i.e., $\prod_{h=1}^H C_h$ is bounded, or strong representation condition (e.g. policy completeness) holds.

## 6  CONCLUSION

While the extant body of provable results in the literature largely focus on *sufficient* conditions for sample-efficient offline RL, this work focuses on obtaining a better understanding of the *necessary* conditions, where we seek to understand to what extent mild assumptions can imply sample-efficient offline RL. This work shows that for off-policy evaluation, even if we are given a representation that can perfectly represent the value function of the given policy and the data distribution has good coverage over the features, any provable algorithm still requires an exponential number of samples to non-trivially approximate the value of the given policy. These results highlight that provable sample-efficient offline RL is simply not possible unless either the distribution shift condition is sufficiently mild or we have stronger representation conditions that go well beyond realizability.

## ACKNOWLEDGMENTS

The authors would like to thank Akshay Krishnamurthy, Alekh Agarwal, Wen Sun, and Nan Jiang for numerous helpful discussion. Sham M. Kakade gratefully acknowledges funding from the ONR award N00014-18-1-2247, and NSF Awards CCF-1703574 and CCF-1740551. Ruosong Wang was supported in part by the NSF IIS1763562, US Army W911NF1920104, and ONR Grant N000141812861. Research performed while Ruosong Wang was an intern at Microsoft Research.

---

[4]Without loss of generality, we can work in a coordinate system such that $\|\theta_h\|_2 \leq H\sqrt{d}$ and $\|\phi(s, a)\|_2 \leq 1$ for all $(s, a) \in \mathcal{S} \times \mathcal{A}$. This follows due to John's theorem (e.g. see (Ball, 1997; Bubeck et al., 2012)).

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

## A  PROOF OF LEMMA 4.2

*Proof.* We first verify $Q^\pi$ is linear for the first $H-1$ levels. For each $(h, c) \in [H-1] \times [\hat{d}]$, we have

$$Q_h^\pi(s_h^c, a_1) = R(s_h^c, a_1) + R(s_{h+1}^{\hat{d}+1}, a_1) + R(s_{h+2}^{\hat{d}+1}, a_1) + \ldots + R(s_H^{\hat{d}+1}, a_1) = r_0 \cdot \hat{d}^{(H-h)/2}.$$

Moreover, for all $a \in \mathcal{A}$,

$$Q_h^\pi(s_h^{\hat{d}+1}, a) = R(s_h^{\hat{d}+1}, a) + R(s_{h+1}^{\hat{d}+1}, a_1) + R(s_{h+2}^{\hat{d}+1}, a_1) + \ldots + R(s_H^{\hat{d}+1}, a_1) = r_0 \cdot \hat{d}^{(H-h+1)/2}.$$

Therefore, if we define

$$\theta_h^\pi = \sum_{c=1}^{\hat{d}} r_0 \cdot \hat{d}^{(H-h)/2} \cdot e_c + \sum_{c=1}^{\hat{d}} Q_h^\pi(s_h^c, a_2) \cdot e_{c+\hat{d}},$$

then $Q_h^\pi(s, a) = (\theta_h^\pi)^\top \phi(s, a)$ for all $(s, a) \in \mathcal{S}_h \times \mathcal{A}$.

Now we verify that the $Q$-function is linear for the last level. Clearly, for all $c \in [\hat{d}]$ and $a \in \mathcal{A}$, $Q_H^\pi(s_H^c, a) = r_0$ and $Q_H^\pi(s_H^{\hat{d}+1}, a) = r_0 \cdot \sqrt{\hat{d}}$. Thus by defining $\theta_H^\pi = \sum_{c=1}^d r_0 \cdot e_c$, we have $Q_H^\pi(s, a) = (\theta_H^\pi)^\top \phi(s, a)$ for all $(s, a) \in \mathcal{S}_H \times \mathcal{A}$.

$\square$

## B  A TECHNICAL LEMMA

We need the following lemma in the proof of our hardness results.

**Lemma B.1.** *Let $\alpha$ be a random variable uniformly distributed on $\{\alpha_+, \alpha_-\}$, where $\alpha_- = 1/2$ and $\alpha_+ = 1/2 + \varepsilon$ with $0 < \varepsilon < 1$. Suppose that $\xi_1, \xi_2, \ldots, \xi_m$ are i.i.d. $\{+1, -1\}$-valued random variables with $\Pr[\xi_i = +1] = \alpha$ for all $i \in [m]$. Let $f$ be a function from $\{+1, -1\}^m$ to $\{\alpha_+, \alpha_-\}$. Suppose $m \le C/\varepsilon^2 \log(1/\delta)$ for some fixed constant $C$. Then*

$$\Pr[f(\xi_1, \xi_2, \ldots, \xi_m) \ne \alpha] > \delta.$$

To our best knowledge, Lemma B.1 was first proved in (Chernoff, 1972) and has enormous applications in statistical learning theory (see, e.g., Chapter 5 in (Anthony & Bartlett, 2009)) and bandits (Mannor & Tsitsiklis, 2004).

To prove Lemma B.1, one can first prove that the maximum likelihood estimator (MLE) is optimal and then show that MLE requires $\Omega(1/\varepsilon^2 \log(1/\delta))$ samples to correctly output $\alpha$ with probability $1 - \delta$ by anti-concentration. Lemma B.1 can also be proved by using information theory. See, e.g., (Kaufmann et al., 2016) for such a proof.

## C  ANOTHER HARD INSTANCE

In this section, we present another hard case under a weaker version of Assumption 1. Here the transition operator is stochastic and the reward is deterministic and sparse, meaning that the reward value is non-zero only for the last level. Moreover, the data distributions $\{\mu_h\}_{h=1}^H$ are induced by a fixed policy $\pi_{\text{data}}$. We also illustrate the construction in Figure 2. Throughout this section, we use $d$ the denote the feature dimension, and we assume $d$ is an even integer for simplicity. We use $\hat{d}$ to denote $d/2 - 1$.

In this section, we adopt the following realizability assumption, which is a weaker version of Assumption 1.

**Assumption 5** (Realizability). *For the policy $\pi : \mathcal{S} \to \Delta(\mathcal{A})$ to be evaluated, there exists $\theta_1, \ldots \theta_H \in \mathbb{R}^d$ such that for all $(s, a) \in \mathcal{S} \times \mathcal{A}$ and $h \in [H]$,*

$$Q_h^\pi(s, a) = \theta_h^\top \phi(s, a).$$

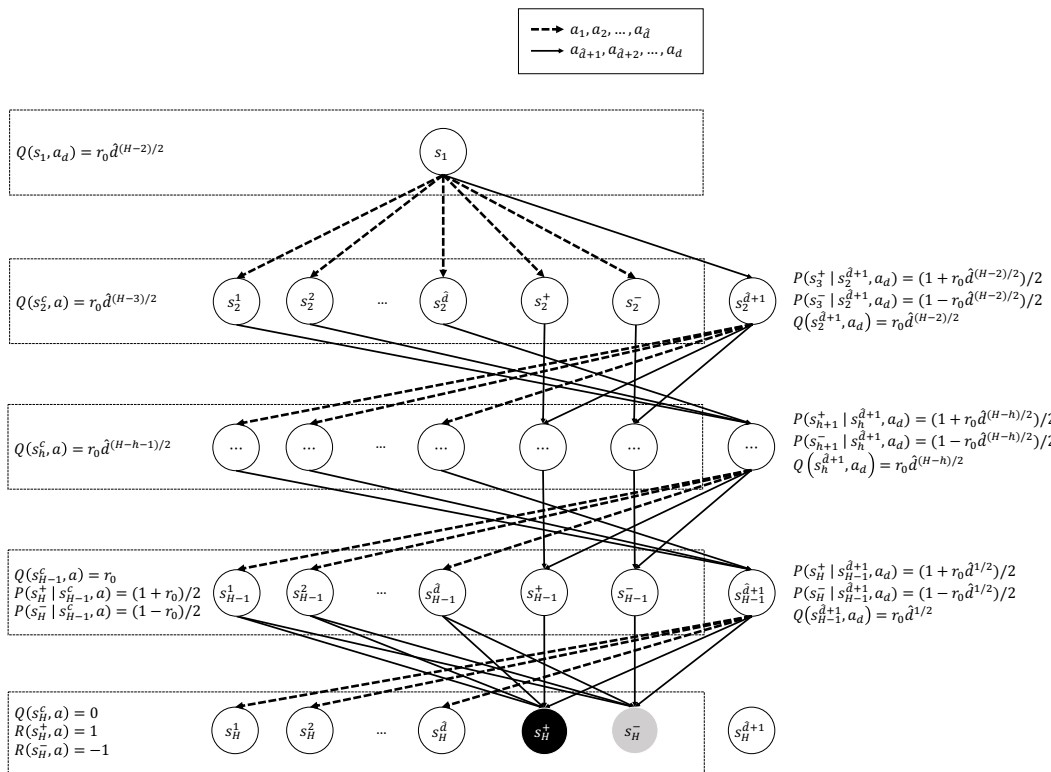

Figure 2: An illustration of the hard instance. Recall that $\hat{d} = d/2 - 1$. States on the top are those in the first level ($h = 1$), while states at the bottom are those in the last level ($h = H$). Dotted line (with arrow) corresponds to transitions associated with actions $a_1, a_2, \ldots, a_{\hat{d}}$, while solid line (with arrow) corresponds to transitions associated with actions $a_{\hat{d}+1}, a_{\hat{d}+2}, \ldots, a_d$. We omit the transition associated with $a_1, a_2, \ldots, a_{\hat{d}}$ in the figure if all actions give the same transition. For each level $h \in [H]$, $Q$-values associated with $s_h^1, s_h^2, \ldots, s_h^{\hat{d}}, s_h^+, s_h^-$ are marked on the left, while transition distributions and $Q$-values associated with $s_h^{\hat{d}+1}$ are marked on the right. Rewards are all deterministic, and the only two states ($s_H^+$ and $s_H^-$) with non-zero reward values are marked in black and grey. Consider the fixed policy that returns $a_d$ for all input states. When $r_0 = 0$, the value of the policy is 0. When $r_0 = \hat{d}^{-(H-2)/2}$, the value of the policy is $= r_0 \hat{d}^{(H-2)/2} = 1$.

**State Space, Action Space and Transition Operator.** In this hard case, the action space $\mathcal{A} = \{a_1, a_2, \ldots, a_d\}$ contains $d$ elements. $\mathcal{S}_1$ contains a single state $s_1$. For each $h \geq 2$, $\mathcal{S}_h$ contains $\hat{d} + 3$ states $s_h^1, s_h^2, \ldots, s_h^{\hat{d}}, s_h^{\hat{d}+1}, s_h^+$ and $s_h^-$.

Let $0 \leq r_0 \leq \hat{d}^{-(H-2)/2}$ be a parameter to be determined. We first define the transition operator for the first level. We have

$$P(s_1, a) = \begin{cases} s_2^c & a = a_c, c \in [\hat{d}] \\ s_2^+ & a = a_{\hat{d}+1} \\ s_2^- & a = a_{\hat{d}+2} \\ s_2^{\hat{d}+1} & a \in \{a_{\hat{d}+3}, a_{\hat{d}+4}, \ldots, a_d\} \end{cases}.$$

Now we define the transition operator when $h \in \{2, 3, \ldots, H-2\}$. For each $h \in \{2, 3, \ldots, H-2\}$, $a \in \mathcal{A}$ and $c \in [\hat{d}]$, we have $P(s_h^c, a) = s_{h+1}^{\hat{d}+1}$, $P(s_h^+, a) = s_{h+1}^+$ and $P(s_h^-, a) = s_{h+1}^-$. For each $h \in \{2, 3, \ldots, H-2\}$ and $c \in [\hat{d}]$, we have $P(s_h^{\hat{d}+1}, a_c) = s_{h+1}^c$. For all $a \in \{a_{\hat{d}+1}, a_{\hat{d}+2}, \ldots, a_d\}$,

we have

$$P\left(s_h^{\hat{d}+1}, a\right) = \begin{cases} s_{h+1}^+ & \text{with probability } (1 + r_0 \cdot \hat{d}^{(H-h)/2})/2 \\ s_{h+1}^- & \text{with probability } (1 - r_0 \cdot \hat{d}^{(H-h)/2})/2 \end{cases}.$$

Now we define the transition operator for the second last level. For all $c \in [\hat{d}]$ and $a \in \mathcal{A}$, we have

$$P(s_{H-1}^c, a) = \begin{cases} s_H^+ & \text{with probability } (1 + r_0)/2 \\ s_H^- & \text{with probability } (1 - r_0)/2 \end{cases}.$$

For all $a \in \mathcal{A}$, we have $P(s_{H-1}^+, a) = s_H^+$ and $P(s_{H-1}^-, a) = s_H^-$. For each $c \in [\hat{d}]$, we have $P(s_{H-1}^{\hat{d}+1}, a_c) = s_H^c$. For all $a \in \{a_{\hat{d}+1}, a_{\hat{d}+2}, \dots, a_d\}$, we have

$$P(s_{H-1}^{\hat{d}+1}, a) = \begin{cases} s_H^+ & \text{with probability } \left(1 + r_0 \cdot \sqrt{\hat{d}}\right)/2 \\ s_H^- & \text{with probability } \left(1 - r_0 \cdot \sqrt{\hat{d}}\right)/2 \end{cases}.$$

**Reward Values.** In this hard case, all reward values are deterministic, and reward values can be non-zero only for the last level. Formally, we have

$$R(s, a) = \begin{cases} 1 & s = s_H^+ \\ -1 & s = s_H^- \\ 0 & \text{otherwise} \end{cases}.$$

**Feature Mapping.** As in the in hard instance in Section 4, let $e_1, e_2, \dots, e_d$ be a set of of orthonormal vectors in $\mathbb{R}^d$. For the initial state, for each $c \in [d]$, we have $\phi(s_1, a_c) = e_c$.

Now we define the feature mapping when $h \in \{2, 3, \dots, H\}$. For each $h \in \{2, 3, \dots, H\}, a \in \mathcal{A}$ and $c \in [\hat{d}], \phi(s_h^c, a) = e_c, \phi(s_h^+, a) = e_{\hat{d}+1}$ and $\phi(s_h^-, a) = e_{\hat{d}+2}$. Moreover, for all actions $a \in \mathcal{A}$,

$$\phi(s_h^{\hat{d}+1}, a) = \begin{cases} e_{\hat{d}+2+c} & a = a_c, c \in [\hat{d}] \\ \frac{1}{\hat{d}^{1/2}}\left(e_1 + e_2 + \dots + e_{\hat{d}}\right) & a \in \{a_{\hat{d}+1}, a_{\hat{d}+2}, \dots, a_d\} \end{cases}.$$

Clearly, for all $(s, a) \in \mathcal{S} \times \mathcal{A}, \|\phi(s, a)\|_2 \le 1$.

**Verifying Assumption 5.** Now we consider the deterministic policy $\pi : \mathcal{S} \to \mathcal{A}$, which is defined to be $\pi(s) = a_d$ for all $s \in \mathcal{S}$. We show that Assumption 5 holds.

When $h = 1$, define

$$\theta_1 = \sum_{c=1}^{\hat{d}} r_0 \cdot \hat{d}^{(H-3)/2} \cdot e_c + e_{\hat{d}+1} - e_{\hat{d}+2} + \sum_{c=1}^{\hat{d}} r_0 \cdot \hat{d}^{(H-2)/2} \cdot e_{\hat{d}+2+c}.$$

For each $h \in \{2, 3, \dots, H-2\}$, define

$$\theta_h = \sum_{c=1}^{\hat{d}} r_0 \cdot \hat{d}^{(H-h-1)/2} \cdot e_c + e_{\hat{d}+1} - e_{\hat{d}+2} + \sum_{c=1}^{\hat{d}} r_0 \cdot \hat{d}^{(H-h-2)/2} \cdot e_{\hat{d}+2+c}.$$

For the second last level $h = H - 1$, define

$$\theta_{H-1} = \sum_{c=1}^{\hat{d}} r_0 \cdot e_c + e_{\hat{d}+1} - e_{\hat{d}+2}.$$

Finally, for the last level $h = H$, define

$$\theta_H = e_{\hat{d}+1} - e_{\hat{d}+2}.$$

It can be verified that for each $h \in [H], Q_h^\pi(s, a) = \theta_h^\top \phi(s, a)$ for all $(s, a) \in \mathcal{S}_h \times \mathcal{A}$.

**The Data Distributions.** For the first level, the data distribution $\mu_1$ is defined to be the uniform distribution over $\{(s_1, a_c) \mid c \in [d]\}$. For each $h \geq 2$, the data distribution $\mu_h$ is a uniform distribution over

$$\{(s_h^1, a_1), (s_h^2, a_1), \ldots, (s_h^{\hat{d}}, a_1), (s_h^+, a_1), (s_h^-, a_1), (s_h^{\hat{d}+1}, a_1), (s_h^{\hat{d}+1}, a_2), \ldots, (s_h^{\hat{d}+1}, a_{\hat{d}})\}.$$

Notice that again $(s_h^{\hat{d}+1}, a)$ is *not* in the support of $\mu_h$ for all actions $a \in \{a_{\hat{d}+1}, a_{\hat{d}+2}, \ldots, a_d\}$. It can be seen that for all $h \in [H]$,

$$\mathbb{E}_{(s,a) \sim \mu_h}[\phi(s,a)\phi(s,a)^\top] = \frac{1}{d}\sum_{c=1}^d e_c e_c^\top = \frac{1}{d}I.$$

Moreover, by defining

$$\pi_{\text{data}}(s) = \begin{cases} \text{Uniform}(\mathcal{A}) & s = s_1 \\ a_1 & s \in \{s_h^c \mid h \in \{2, 3, \ldots, H\}, c \in [\hat{d}]\} \\ a_1 & s \in \{s_h^+ \mid h \in \{2, 3, \ldots, H\}\} \\ a_1 & s \in \{s_h^- \mid h \in \{2, 3, \ldots, H\}\} \\ \text{Uniform}(\{a_1, a_2, \ldots, a_{\hat{d}}\}) & s \in \{s_h^{\hat{d}+1} \mid h \in \{2, 3, \ldots, H\}\} \end{cases},$$

we have $\mu_h = \mu_h^{\pi_{\text{data}}}$ for all $h \in [H]$.

**The Lower Bound.** Now we show that it is information-theoretically hard for any algorithm to distinguish the case $r_0 = 0$ and $r_0 = \hat{d}^{-(H-2)/2}$ in the offline setting by taking samples from the data distributions $\{\mu_h\}_{h=1}^H$. Here we consider the above policy $\pi$ defined above which returns action $a_d$ for all input states. Notice that when $r_0 = 0$, the value of the policy would be zero. On the other hand, when $r_0 = \hat{d}^{-(H-2)/2}$, the value of the policy would be $r_0 \cdot \hat{d}^{(H-2)/2} = 1$. Therefore, if the algorithm approximates the value of the policy up to an approximation error of $1/2$, then it must distinguish the case that $r_0 = 0$ and $r_0 = \hat{d}^{-(H-2)/2}$.

We first notice that for the case $r_0 = 0$ and $r_0 = \hat{d}^{-(H-2)/2}$, the data distributions $\{\mu_h\}_{h=1}^H$, the feature mapping $\phi : \mathcal{S} \times \mathcal{A} \to \mathbb{R}^d$, the policy $\pi$ to be evaluated and the reward distributions $R$ are the same. Thus, in order to distinguish the case $r_0 = 0$ and $r_0 = \hat{d}^{-(H-2)/2}$, the only way is to query the transition operator $P$ by using sampling taken from the data distributions.

Now, for all state-action pairs $(s, a)$ in the support of the data distributions of the first $H - 2$ levels (namely $\mu_1, \mu_2, \ldots, \mu_{H-2}$), the transition operator will be identical. This is because changing $r_0$ only changes the transition distributions of $(s_h^{\hat{d}+1}, a_{\hat{d}+1}), (s_h^{\hat{d}+1}, a_{\hat{d}+2}), \ldots, (s_h^{\hat{d}+1}, a_d)$, and such state-actions are not in the support of $\mu_h$ for all $h \in [H - 2]$. Moreover, for any $(s, a) \in \{s_{H-1}^+, s_{H-1}^-, s_{H-1}^{\hat{d}+1}\} \times \mathcal{A}$ in the support of $\mu_{H-1}$, $P(s, a)$ will also be identical no matter $r_0 = 0$ or $r_0 = \hat{d}^{-(H-2)/2}$. For those state-action pairs $(s, a)$ in the support of $\mu_{H-1}$ with $s \notin \{s_{H-1}^+, s_{H-1}^-, s_{H-1}^{\hat{d}+1}\}$, we have

$$P(s, a) = \begin{cases} s_H^+ & \text{with probability } (1 + r_0)/2 \\ s_H^- & \text{with probability } (1 - r_0)/2 \end{cases}.$$

Again, this is because $(s_{H-1}^{\hat{d}+1}, a)$ is not in the support of $\mu_{H-1}$ for all $a \in \{a_{\hat{d}+1}, a_{\hat{d}+2}, \ldots, a_d\}$.

Therefore, in order to distinguish the case $r_0 = 0$ and $r_0 = \hat{d}^{-(H-2)/2}$, the agent needs distinguish two transition distributions

$$p_1 = \begin{cases} s_H^+ & \text{with probability } 1/2 \\ s_H^- & \text{with probability } 1/2 \end{cases}$$

and

$$p_2 = \begin{cases} s_H^+ & \text{with probability } (1 + \hat{d}^{-(H-2)/2})/2 \\ s_H^- & \text{with probability } (1 - \hat{d}^{-(H-2)/2})/2 \end{cases}.$$

Again, by Lemma B.1, in order to distinguish $p_1$ and $p_2$ with probability at least $0.9$, one needs $\Omega(\hat{d}^{H-2})$ samples. Formally, we have the following theorem.

**Theorem C.1.** *Suppose Assumption 2 holds, and rewards are deterministic and could be none-zero only for state-action pairs in the last level. Fix an algorithm that takes as input both a policy and a feature mapping. There exists an MDP satisfying Assumption 5, such that for a fixed policy $\pi : \mathcal{S} \to \mathcal{A}$, the algorithm requires $\Omega((d/2 - 1)^{H/2})$ samples to output the value of $\pi$ up to constant additive approximation error with probability at least $0.9$.*

## D   ANALYSIS OF ALGORITHM 1

### D.1   PROOF OF LEMMA 5.1

Clearly,

$$
\hat{\theta}_h = \hat{\Lambda}_h^{-1} \left( \sum_{i=1}^{N} \phi(s_h^i, a_h^i) \cdot (r_h^i + \hat{V}_{h+1}(\bar{s}_h^i)) \right)
$$

$$
= \hat{\Lambda}_h^{-1} \left( \sum_{i=1}^{N} \phi(s_h^i, a_h^i) \cdot (r_h^i + \hat{Q}_{h+1}(\bar{s}_h^i, \pi(\bar{s}_h^i))) \right)
$$

$$
= \hat{\Lambda}_h^{-1} \left( \sum_{i=1}^{N} \phi(s_h^i, a_h^i) \cdot (r_h^i + \phi(\bar{s}_h^i, \pi(\bar{s}_h^i))^\top \hat{\theta}_{h+1}) \right)
$$

$$
= \hat{\Lambda}_h^{-1} \left( \sum_{i=1}^{N} \phi(s_h^i, a_h^i) \cdot (r_h^i + \phi(\bar{s}_h^i, \pi(\bar{s}_h^i))^\top \theta_{h+1}) + \sum_{i=1}^{N} \phi(s_h^i, a_h^i) \cdot \phi(\bar{s}_h^i, \pi(\bar{s}_h^i))^\top (\hat{\theta}_{h+1} - \theta_{h+1}) \right)
$$

$$
= \hat{\Lambda}_h^{-1} \left( \sum_{i=1}^{N} \phi(s_h^i, a_h^i) \cdot (r_h^i + \phi(\bar{s}_h^i, \pi(\bar{s}_h^i))^\top \theta_{h+1}) \right) + \hat{\Lambda}_h^{-1} \left( \sum_{i=1}^{N} \phi(s_h^i, a_h^i) \cdot \phi(\bar{s}_h^i, \pi(\bar{s}_h^i))^\top (\hat{\theta}_{h+1} - \theta_{h+1}) \right).
$$

For the first term, we have

$$
\hat{\Lambda}_h^{-1} \left( \sum_{i=1}^{N} \phi(s_h^i, a_h^i) \cdot (r_h^i + \phi(\bar{s}_h^i, \pi(\bar{s}_h^i))^\top \theta_{h+1}) \right)
$$

$$
= \hat{\Lambda}_h^{-1} \left( \sum_{i=1}^{N} \phi(s_h^i, a_h^i) \cdot (r_h^i + Q^\pi(\bar{s}_h^i, \pi(\bar{s}_h^i))) \right)
$$

$$
= \hat{\Lambda}_h^{-1} \left( \sum_{i=1}^{N} \phi(s_h^i, a_h^i) \cdot (r_h^i + V^\pi(\bar{s}_h^i)) \right)
$$

$$
= \hat{\Lambda}_h^{-1} \left( \sum_{i=1}^{N} \phi(s_h^i, a_h^i) \cdot (Q^\pi(s_h^i, a_h^i) + \xi_h^i) \right)
$$

$$
= \hat{\Lambda}_h^{-1} \sum_{i=1}^{N} \phi(s_h^i, a_h^i) \cdot \xi_h^i + \hat{\Lambda}_h^{-1} \sum_{i=1}^{N} \phi(s_h^i, a_h^i) \cdot \phi(s_h^i, a_h^i)^\top \theta_h
$$

$$
= \hat{\Lambda}_h^{-1} \sum_{i=1}^{N} \phi(s_h^i, a_h^i) \cdot \xi_h^i + \hat{\Lambda}_h^{-1} (\Phi_h^\top \Phi_h) \theta_h
$$

$$
= \hat{\Lambda}_h^{-1} \Phi_h \xi_h + \theta_h - \lambda \hat{\Lambda}_h^{-1} \theta_h.
$$

Therefore,

$$
\hat{\theta}_1 - \theta_1 = (\hat{\Lambda}_1^{-1} \Phi_1 \xi_1 - \lambda \hat{\Lambda}_1^{-1} \theta_1) + \hat{\Lambda}_1^{-1} \Phi_1^\top \overline{\Phi}_2 (\theta_2 - \hat{\theta}_2)
$$

$$
= (\hat{\Lambda}_1^{-1} \Phi_1 \xi_1 - \lambda \hat{\Lambda}_1^{-1} \theta_1) + \hat{\Lambda}_1^{-1} \Phi_1^\top \overline{\Phi}_2 (\hat{\Lambda}_2^{-1} \Phi_2^\top \xi_2 - \lambda \hat{\Lambda}_2^{-1} \theta_2)
$$

$$
+ \hat{\Lambda}_1^{-1} \Phi_1^\top \overline{\Phi}_2 \hat{\Lambda}_2^{-1} \Phi_2^\top \overline{\Phi}_3 (\theta_3 - \hat{\theta}_3)
$$

$$
= \ldots
$$

$$
= \sum_{h=1}^{H} \hat{\Lambda}_1^{-1} \Phi_1^\top \overline{\Phi}_2 \hat{\Lambda}_2^{-1} \Phi_2^\top \overline{\Phi}_3 \cdots (\hat{\Lambda}_h^{-1} \Phi_h^\top \xi_h - \lambda \hat{\Lambda}_h^{-1} \theta_h).
$$

Also note that

$$(Q^\pi(s_1, \pi(s_1)) - \hat{Q}(s_1, \pi(s_1)))^2 = \|\theta_1 - \hat{\theta}_1\|^2_{\Lambda_1}.$$

## D.2 PROOF OF THEOREM 5.2

By matrix concentration inequality (Tropp, 2015), we have the following lemma.

**Lemma D.1.** *For each $h \in [H]$, with probability $1 - \delta/(4H)$, for some universal constant $C$, we have*

$$\left\| \frac{1}{N} \Phi_h^\top \Phi_h - \Lambda_h \right\|_2 \le C\sqrt{d\log(dH/\delta)/N}.$$

*and*

$$\left\| \frac{1}{N} \overline{\Phi}_{h+1} \overline{\Phi}_{h+1} - \overline{\Lambda}_{h+1} \right\|_2 \le C\sqrt{d\log(dH/\delta)/N}.$$

*Therefore, since $\lambda = CH\sqrt{d\log(dH/\delta)N}$, with probability $1 - \delta/(4H)$, we have*

$$\hat{\Lambda}_h = \Phi_h^\top \Phi_h + \lambda I \succeq N\Lambda_h.$$

Note that

$$(Q^\pi(s_1, \pi(s_1)) - \hat{Q}(s_1, \pi(s_1)))^2$$

$$\le H \cdot \left( \sum_{h=1}^{H} \left\| \hat{\Lambda}_1^{-1} \Phi_1^\top \overline{\Phi}_2 \hat{\Lambda}_2^{-1} \Phi_2^\top \overline{\Phi}_3 \cdots (\hat{\Lambda}_h^{-1} \Phi_h^\top \xi_h - \lambda \hat{\Lambda}_h^{-1} \theta_h) \right\|^2_{\overline{\Lambda}_1} \right)$$

$$\le 2H \cdot \left( \sum_{h=1}^{H} \left\| \hat{\Lambda}_1^{-1} \Phi_1^\top \overline{\Phi}_2 \hat{\Lambda}_2^{-1} \Phi_2^\top \overline{\Phi}_3 \cdots \hat{\Lambda}_h^{-1} \Phi_h^\top \xi_h \right\|^2_{\overline{\Lambda}_1} + \sum_{h=1}^{H} \left\| \hat{\Lambda}_1^{-1} \Phi_1^\top \overline{\Phi}_2 \hat{\Lambda}_2^{-1} \Phi_2^\top \overline{\Phi}_3 \cdots \lambda \hat{\Lambda}_h^{-1} \theta_h \right\|^2_{\overline{\Lambda}_1} \right).$$

For each $h \in [H]$,

$$\|\hat{\Lambda}_1^{-1} \Phi_1^\top \overline{\Phi}_2 \hat{\Lambda}_2^{-1} \Phi_2^\top \overline{\Phi}_3 \cdots \hat{\Lambda}_h^{-1} \Phi_h^\top \xi_h\|^2_{\overline{\Lambda}_1}$$

$$\le \|\Phi_1 \hat{\Lambda}_1^{-1} \overline{\Lambda}_1 \hat{\Lambda}_1^{-1} \Phi_1^\top\|_2 \cdot \|\overline{\Phi}_2 \hat{\Lambda}_2^{-1} \Phi_2^\top \overline{\Phi}_3 \cdots \hat{\Lambda}_h^{-1} \Phi_h^\top \xi_h\|^2_2$$

$$\le \|\hat{\Lambda}_1^{-1/2} \overline{\Lambda}_1 \hat{\Lambda}_1^{-1/2}\|_2 \cdot \|\Phi_1 \hat{\Lambda}_1^{-1} \Phi_1^\top\|_2 \cdot \|\overline{\Phi}_2 \hat{\Lambda}_2^{-1} \Phi_2^\top \overline{\Phi}_2 \cdots \hat{\Lambda}_h^{-1} \Phi_h^\top \xi_h\|^2_2$$

$$\le \|\hat{\Lambda}_1^{-1/2} \overline{\Lambda}_1 \hat{\Lambda}_1^{-1/2}\|_2 \cdot \prod_{h'=1}^{h-1} \left( \|\Phi_{h'} \hat{\Lambda}_{h'}^{-1} \Phi_{h'}^\top\|_2 \cdot \|\hat{\Lambda}_{h'+1}^{-1/2} (\overline{\Phi}_{h'+1}^\top \overline{\Phi}_{h'+1}) \hat{\Lambda}_{h'+1}^{-1/2}\|_2 \right) \cdot \|\xi_h\|^2_{\Phi_h \hat{\Lambda}_h^{-1} \Phi_h^\top}.$$

Similarly,

$$\|\hat{\Lambda}_1^{-1} \Phi_1^\top \overline{\Phi}_2 \hat{\Lambda}_2^{-1} \Phi_2^\top \overline{\Phi}_3 \cdots \lambda \hat{\Lambda}_h^{-1} \theta_h\|^2_{\overline{\Lambda}_1}$$

$$\le \|\hat{\Lambda}_1^{-1/2} \overline{\Lambda}_1 \hat{\Lambda}_1^{-1/2}\|_2 \cdot \prod_{h'=1}^{h-1} \left( \|\Phi_{h'} \hat{\Lambda}_{h'}^{-1} \Phi_{h'}^\top\|_2 \cdot \|\hat{\Lambda}_{h'+1}^{-1/2} (\overline{\Phi}_{h'+1}^\top \overline{\Phi}_{h'+1}) \hat{\Lambda}_{h'+1}^{-1/2}\|_2 \right) \cdot \lambda^2 \cdot \|\theta_h\|^2_{\hat{\Lambda}_h^{-1}}$$

$$\le \|\hat{\Lambda}_1^{-1/2} \overline{\Lambda}_1 \hat{\Lambda}_1^{-1/2}\|_2 \cdot \prod_{h'=1}^{h-1} \left( \|\Phi_{h'} \hat{\Lambda}_{h'}^{-1} \Phi_{h'}^\top\|_2 \cdot \|\hat{\Lambda}_{h'+1}^{-1/2} (\overline{\Phi}_{h'+1}^\top \overline{\Phi}_{h'+1}) \hat{\Lambda}_{h'+1}^{-1/2}\|_2 \right) \cdot \lambda \cdot H^2 d.$$

For all $h \in [H]$, we have

$$\|\Phi_h \hat{\Lambda}_h^{-1} \Phi_h^\top\|_2 \le 1$$

and

$$\|\hat{\Lambda}_h^{-1/2} (\overline{\Phi}_h^\top \overline{\Phi}_h) \hat{\Lambda}_h^{-1/2}\|_2 \le \|N\hat{\Lambda}_h^{-1/2} \overline{\Lambda}_h \hat{\Lambda}_h^{-1/2}\|_2 + \|\hat{\Lambda}_h^{-1/2} (\overline{\Phi}_h^\top \overline{\Phi}_h - N\overline{\Lambda}_h) \hat{\Lambda}_h^{-1/2}\|_2.$$

Conditioned on the event in Lemma D.1,

$$\hat{\Lambda}_h \succeq N\Lambda_h \succeq \frac{N}{C_h} \overline{\Lambda}_h,$$

which implies $\|N\hat{\Lambda}_h^{-1/2}\overline{\Lambda}_h\hat{\Lambda}_h^{-1/2}\| \leq C_h$. Moreover, conditioned on the event in Lemma D.1,

$$\|\hat{\Lambda}_h^{-1/2}(\overline{\Phi}_h^\top\overline{\Phi}_h - N\overline{\Lambda}_h)\hat{\Lambda}_h^{-1/2}\|_2 \leq C\sqrt{d\log(dH/\delta)N}/\lambda.$$

Thus,

$$\|\hat{\Lambda}_1^{-1/2}\overline{\Lambda}_1\hat{\Lambda}_1^{-1/2}\|_2 \leq C_1/N.$$

and

$$\|\hat{\Lambda}_h^{-1/2}(\overline{\Phi}_h^\top\overline{\Phi}_h)\hat{\Lambda}_h^{-1/2}\|_2 \leq C_h + C\sqrt{d\log(dH/\delta)N}/\lambda.$$

Finally, by Theorem 1.2 in (Hsu et al., 2012a), with probability $1 - \delta/(4H)$, for some constant $C'$, we have

$$\|\xi_h\|^2_{\Phi_h\hat{\Lambda}_h^{-1}\Phi_h^\top} \leq C'H^2 d\log(H/\delta).$$

Therefore,

$$\left\|\hat{\Lambda}_1^{-1}\Phi_1^\top\overline{\Phi}_2\hat{\Lambda}_2^{-1}\Phi_2^\top\overline{\Phi}_3\cdots\hat{\Lambda}_h^{-1}\Phi_h^\top\xi_h\right\|^2_{\overline{\Lambda}_1} + \left\|\hat{\Lambda}_1^{-1}\Phi_1^\top\overline{\Phi}_2\hat{\Lambda}_2^{-1}\Phi_2^\top\overline{\Phi}_3\cdots\lambda\hat{\Lambda}_h^{-1}\theta_h\right\|^2_{\overline{\Lambda}_1}$$
$$\leq\frac{C_1}{N}(C_2 + C\sqrt{d\log(d/\delta)N}/\lambda)\times\cdots\times(C_h + C\sqrt{d\log(d/\delta)N}/\lambda)\times(C'H^2 d\log(H/\delta) + \lambda H^2 d)$$
$$\leq\frac{C_1}{N}(C_2 + 1/H)\times\cdots\times(C_h + 1/H)\times(C'H^2 d\log(H/\delta) + \lambda H^2 d)$$
$$\leq\frac{e}{N}C_1 \times C_2 \times \cdots \times C_h \times (C'H^2 d\log(H/\delta) + CdH^3\sqrt{d\log(dH/\delta)N}).$$

Let $c > 0$ be a large enough constant. We now have

$$\mathbb{E}_{s_1}[(Q_1^\pi(s_1,\pi(s_1)) - \hat{Q}_1(s_1,\pi(s_1)))^2] \leq c\cdot\left(\prod_{h=1}^H C_h\right)\cdot dH^5\cdot\sqrt{\frac{d\log(dH/\delta)}{N}}.$$

