# OpenReview forum: "What are the Statistical Limits of Offline RL with Linear Function Approximation?"
_ICLR.cc/2021/Conference — ICLR 2021 Spotlight_

### Official Review · AnonReviewer4 · 2020-10-27

**Rating:** 7
**Confidence:** 4

**Review:**

It's an interesting paper showing provably efficient batch RL is impossible when only two assumptions holding: a) Realizability, and b) Feature Coverage. By providing counter-examples, which is quite persuasive as the State/Action Space is only polynomially dependent in H, the sample complexity could be exponentially dependent in H w.h.p.

However, I still have doubts about the Assumption Feature Coverage. Different than Assumption Concentrability [Chen & Jiang(2019), Xie & Jiang(2020a)], Feature Coverage only assumes that feature covariance matrix has minimum eigenvalue 1/d. My doubts then follow:
1) Is Asp Concentrability a more stringent asp than Asp Feature Coverage (from a mathematical perspective)? If so, why we would concern Asp Feature Coverage while Chen & Jiang(2019) already proves Asp Concentrability is necessary for efficient batch RL?
2) Can Asp Feature Coverage really represent exploratory of the batch dataset? In RL's perspective, exploratory is important empirically and theoretically. However, in the hard instance, the important states $s_h^*$ are even not in the support of the dataset (maybe also breaks the Asp Concentrability?), though I understand it's necessary to obtain the exponential lower bound.
3) Theorem 5.1 confuses me a little. I understand the main goal is to illustrate Error Amplification. But why we would concern upper bound instead of the lower bound? Or Sec5 aims to say that Error Amplification is unavoidable under Realizability and Feature Coverage?

In conclusion, I argue that this paper is well-written and answers the question perfectly with certain assumptions. I have also checked the proof and there are no significant errors. Though I have some doubts about whether certain assumptions are reasonable enough in RL, I still stay positive about this paper. My doubts may be incorrect, but I'm looking forward to the discussion.

---

> ### Author Response · Authors · 2020-11-20
> **Response**
>
> **Is Asp Concentrability a more stringent asp than Asp Feature Coverage (from a mathematical perspective)? If so, why we would concern Asp Feature Coverage while Chen & Jiang(2019) already proves Asp Concentrability is necessary for efficient batch RL?**
>
> We agree that Assumption Concentrability is more stringent than Assumption Feature Coverage. However, we disagree that Chen & Jiang (2019) have proved that Assumption Concentrability is necessary for efficient batch RL in the linear function approximation setting considered in this paper. Chen & Jiang (2019) considered general function classes, and to prove the necessity of Assumption Concentrability, Chen & Jiang (2019) used an unstructured function class which cannot be represented by linear functions of low-dimensional features. Since we focus on linear functions in this paper, such a lower bound does not apply in our setting.
>
> **Can Asp Feature Coverage really represent exploratory of the batch dataset?**
>
> The concentrability coefficient is the largest possible ratio between the probability for a state-action pair $(s, a)$ to be visited by a policy, and the probability that $(s, a)$ appears on the data distribution. In this paper, we measure the distribution shift in terms of the spectrum of the feature covariance matrix for the following reasons.
>
> 1. Concentratability coefficients consider worst-case density ratio among all state-action pairs. However, in the linear function approximation setting where an additional feature mapping is adopted for generalization, one should take the feature mapping into consideration. E.g., state-actions with similar features should be considered similar. On the other hand, the definition of concentrability coefficient is totally irrelevant to the feature mapping. Moreover, when the state space is huge or even continuous, the concentrability coefficient could also be huge or even unbounded, no matter which data distribution is used.
>
> 2. Since we focus on linear function approximation in this paper, we should consider distribution shift conditions that are meaningful for linear functions. For the case of linear regression (which is equivalent to offline policy evaluation with $H=1$), low distribution shift in terms of the spectrum of the feature covariance matrix is a well-known sufficient condition in the context of supervised learning.
>
> Finally, measuring the distribution shift in terms of the spectrum of the feature covariance matrix is also adopted in prior work on offline policy evaluation with linear function approximation. E.g., Duan & Wang (2020) (cited in the updated version) showed that under stronger representation conditions, low distribution shift in terms of the spectrum of the feature covariance matrix is sufficient for sample efficient offline policy evaluation.
>
> **The important states are even not in the support of the dataset**
>
> We would like to remind the reviewer that in our hardness instance, the size of the state space is only $O(dH)$. With such a small state space, if all important states are in the support of the dataset, then the instance will basically become a tabular case where all state-action pairs are visited with sufficiently high probability. In fact, the whole point of using function approximation is to allow the agent to generalize knowledge learned from other states to (unknown) important states, to deal with possibly huge or even continuous state spaces. Moreover, one can definitely increase the size of the state space in our hard instance and change the data distribution so that important states are visited with low (exponentially small) probability, in which case the lower bound still holds. However, we decide to keep the current construction for simplicity and clarity.
>
> **But why we would concern upper bound instead of the lower bound?**
>
> In Section 4 we have formally proved that *any* algorithm requires an exponential number of samples to estimate the value of a given policy, which formalizes the issue of error amplification in batch RL. The goal of Section 5 is two-fold. First, we provide sufficient conditions (low distribution shift and strong representation conditions) that enable sample efficient batch RL, to support the claim that "Our results highlight that sample-efficient, batch RL is not guaranteed unless significantly stronger conditions, such as the distribution shift is sufficiently mild or representation conditions that are far stronger than realizability, are met". On the other hand, in the paragraph following Theorem 5.1, we demonstrate how the issue of error amplification arise when running specific algorithms (LSVI, called LSPE in the updated version) on our hard instance to better illustrate the issue of error amplification.

---

> > ### Comment · AnonReviewer4 · 2020-11-25
> > **Response**
> >
> > Thank you for the detailed response. Based on the feedback, I plan to raise my score to 7.

---

### Official Review · AnonReviewer1 · 2020-10-27
**Sound theoretical analysis**

**Rating:** 8
**Confidence:** 3

**Review:**

The paper provides a theoretical analysis on the sample complexity of OPE with linear function approximation and assumptions on representation and distribution shift. The main results shows that realizability and feature coverage are not sufficient to guarantee a polynomial sample complexity. The paper also provides a performance guarantee for LSVI with a stronger assumption on the distribution shift.

In general, the paper provides a sound (and correct) theoretical analysis, the construction of the hard instance is new. In section 5,  I think the measure of the distribution shift in terms of the covariance matrix is also new. I believe the paper can have a significant contribution to the batch RL/offline RL community. Therefore, I recommend to accept this paper.

##### Comments and clarification questions
In terms of clarity, the main text is clearly written, and the related work section is comprehensive.

The paper mentions that ”batch RL is not guaranteed unless stronger conditions on the distribution shift or stronger representation conditions beyond realizability”. However, the paper seems to focus on the former and does not have any analysis for the latter in the main text. For example, what is the guarantee of LSVI under closeness under Bellman update (without mild distribution shift assumption)?

The paper might benefit from giving some explanations on why low C_h (defined in assumption 3) implies mild distribution shift.  Moreover, can you elaborate more on this sentence "we measure the distribution shift in terms of the spectrum of the feature covariance matrix of the data distribution, which is more natural than the concentrability coefficient for the case of linear function approximation"? What is the connection between assumption 3 and the concentrability coefficient?

I did not carefully check the proof in the appendix, therefore, I give myself a slightly low confidence score.

**
Post-rebuttal update: the score is increased from 7 to 8.
**

---

> ### Author Response · Authors · 2020-11-20
> **Response**
>
> **The paper seems to focus on the former and does not have any analysis for the latter in the main text**
>
> In Section 5 of the updated version, we provide an *equality* to characterize the estimation error of LSVI (called LSPE in the updated version). Then we discuss two cases when the estimation error is low: (1) low distribution shift and (2) strong representation condition. In the previous version, we mainly focused on the former since previous work has provided near-optimal bound for LSVI under strong representation conditions. For instance, Duan & Wang (2020) proved that under Assumption 2 (Feature Coverage) and closeness under Bellman update, LSVI achieves polynomial sample complexity for offline policy evaluation with linear function approximation. Nevertheless, in the updated version we have provided more discussion to make this clear.
>
> **Why low C_h (defined in assumption 3) implies mild distribution shift**
>
> To see why low $C_h$ in Assumption 3 implies low distribution shift, consider the case where the behavior policy (the policy that induces the data distributions) is exactly the same as the target policy (the policy to be evaluated), i.e., no distribution shift. In this case, $C_h = 1$ for all $h$, and thus the product of all $C_h$ is bounded. When the behavior policy is different from the target policy, $C_h$ will be larger than $1$. In general, $C_h$ characterizes the difference between the data distribution and the one-step lookahead distribution induced by the data distribution and the target policy, in a spectral sense. Finally, $C_h$ arises naturally as one applies Cauchy-Schwarz inequality to break the product of matrices in Equation (1) into contributions from different levels.
>
> **Can you elaborate more on this sentence "we measure the distribution shift in terms of the spectrum of the feature covariance matrix of the data distribution, which is more natural than the concentrability coefficient for the case of linear function approximation"?**
>
> The concentrability coefficient is the largest possible ratio between the probability for a state-action pair $(s, a)$ to be visited by a policy, and the probability that $(s, a)$ appears on the data distribution. In this paper, we measure the distribution shift in terms of the spectrum of the feature covariance matrix for the following reasons.
>
> 1. Concentratability coefficients consider worst-case density ratio among all state-action pairs. However, in the linear function approximation setting where an additional feature mapping is adopted for generalization, one should take the feature mapping into consideration. E.g., state-actions with similar features should be considered similar. On the other hand, the definition of concentrability coefficient is totally irrelevant to the feature mapping. Moreover, when the state space is huge or even continuous, the concentrability coefficient could also be huge or even unbounded, no matter which data distribution is used.
>
> 2. Since we focus on linear function approximation in this paper, we should consider distribution shift conditions that are meaningful for linear functions. For the case of linear regression (which is equivalent to offline policy evaluation with $H=1$), low distribution shift in terms of the spectrum of the feature covariance matrix is a well-known sufficient condition in the context of supervised learning.
>
> Finally, measuring the distribution shift in terms of the spectrum of the feature covariance matrix is also adopted in prior work on offline policy evaluation with linear function approximation. E.g., Duan & Wang (2020) (cited in the updated version) showed that under stronger representation conditions, low distribution shift in terms of the spectrum of the feature covariance matrix is sufficient for sample efficient offline policy evaluation.

---

> > ### Comment · AnonReviewer1 · 2020-11-24
> > **Short response**
> >
> > Thank you for writing the detailed response. Based on the authors' response, I think it is a good paper so I increase the score to 8.

---

### Official Review · AnonReviewer3 · 2020-10-28
**Important demonstration that off-policy RL requires a number of samples exponential in the horizon of the MDP for accurate value function approximation**

**Rating:** 7
**Confidence:** 3

**Review:**

### Summary

This is a theoretical paper proving an MDP exists in which there is a lower bound on the number of off-policy samples necessary to sufficiently linearly approximate the Q-functions which is exponential in the time horizon of the MDP.  In addition, they demonstrate that while this can be mitigated with samples pulled from a similar policy to the optimal one, the sample distribution must be quite similar, due to cascading geometric error.  The paper is a challenging one to read, and I can't say I followed the whole proofs, but I understand the results.

### Significance

This result is quite important in understanding the feasibility of off-policy approximation.  While the proofs are on linear approximation schemes, it provides intuition into the behaviors of more modern off-policy non-linear schemes, as well.  Of course we intuitively knew off-policy approximation becomes more difficult the more dissimilar the samples are from the optimal policy, but the theoretical support of exponential sample size is very useful and meaningful.

### Originality

The paper's results are new, to my experience.

### Quality and Clarity

The proofs are creative and well done.  I do wish the intuitive explanations of the theorems were better organized and more fleshed out.  There is a heavy dependence on citation.  For example, "It is well known that in order to distinguish r_1 and r_2 with probability at least 0.9, any algorithm requires <many> samples." fills a very key place in understanding the theorem.  I'm not familiar with this result, and without it, I don't understand a key moment of the proof.

### Conclusion

I think it's an important paper, and is highly relevant to the large amount of work that uses off-policy samples.  I don't think it's written very clearly, which is a shame.

---

> ### Author Response · Authors · 2020-11-20
> **Response**
>
> In order to make the proof more readable, we have provided a technical lemma in Appendix B of the updated version. In Lemma B.1, we provide a formal statement of the hardness result appeared in prior work. We discussed how to invoke Lemma B.1 to finish the proof of our hardness result and also discussed the intuition behind Lemma B.1.
>
> We are happy to add further clarification/intuition in the next version if the reviewer still finds anything unclear. Meanwhile, we would like to stress that to demonstrate the issue of error amplification, the sophisticated construction is indeed necessary for proving our hardness results.

---

### Official Review · AnonReviewer2 · 2020-10-30
**A good constructive example of the impossibility of efficient Batch RL with realizability and uncorrelated features.**

**Rating:** 8
**Confidence:** 3

**Review:**

This paper presents a impossibility result for value-function approximation in batch-mode RL. The chart below puts this work in context. This work essentially shows -- through a constructive example of an MDP -- that the amount of data needed for approximating Q values must increases exponentially with the horizon in episodic RL tasks even if we assume that the Q-values are realizable and that the features gathered by the behavior policy -- that collected the data -- are uncorrelated. This problem arises because the data gathering policy can fail to get data from all states even though the features themselves are uncorrelated.

The authors claim that this setting is interesting because these conditions only require polynomial-number of samples in the standard supervised learning case, and I agree with that. Broadly I checked the main construction in the paper and the result seems correct. I think this paper will be a good addition to ICLR.

      |                | Strong Concentratibility|
      |                | extra conditions on     | (Xie Jiang 2020)
      |                |  the dynamics of the MDP|  [✓]
      |                |
      |Coverage of the | Concentrability         |       Szepesvári and Munos 2005
      |behavior policy | (experience gathered    |                      [✓]
      |                |  from every state)      |
      |                |
      |                | Uncorrelated            | This work
      |                | Features                | [x]
      |________________|______________________________________________________________
      |                                           Realizability     Bellman Closedness
      | bellman closed = Every intermediate value function arrived at after
      | bellman update is also linearly realizable
      |_______________________________________________________________________________
      |                             Representability of Q values


## Some comments

1. Assumption 2 will actually imply that  σ_max = 1/d as well because (σ_min + ... + σ_max) ≥ d σ_min = 1 and sum of eigenvalues equals the trace, and exchanging trace and expectation will give (σ_min + ... + σ_max) ≤ 1. I.e. the covariance is necessarily scaled identity matrix ⇒ the features are uncorrelated.

---

> ### Author Response · Authors · 2020-11-20
> **Response**
>
> We would like to thank the reviewer for the positive review. Indeed Assumption 2 implies $\sigma_{\mathrm{max}} = \sigma_{\mathrm{min}} = 1/d$. We have provided more discussion in Assumption 2 of the updated version.

---

### Author Response · Authors · 2020-11-20
**Paper Revision**

We thank all reviewers for their comments and suggestions! We have revised our paper accordingly. The main changes are summarized as follows.

1. We have strengthened our main hardness result. In the updated version, we show that for our hard instance, the $Q$-functions of *all* policies are linear with respect to the given feature mapping (see Assumption 1), and evaluating *any* single policy requires an exponential number of samples. Our new representation condition is significantly stronger than assuming realizability with regards to a single target policy; it assumes realizability for all policies. Regardless, even under this stronger representation condition, it is hard to evaluate any policy. See the updated version for more details.

2. We have added Remark 3, which shows an interesting separation between Least-Squares Value Iteration (LSVI) and Least-Squares Policy Iteration (LSPI) implied by our hardness result.

3. We have reorganized Section 5. In the updated version, we begin the section by introducing an *equality* that characterizes the estimation error of LSVI (called LSPE in the updated version). Then we discuss two cases when the estimation error is low: (1) low distribution shift and (2) strong representation condition.

In the meantime, we are happy to answer any questions or concerns that remain. We hope to hear back from the reviewers soon!

---

### Decision · Program_Chairs · 2021-01-07
**Final Decision**

**Decision:**

Accept (Spotlight)

**Comment:**

The paper shows hardness results for batch reinforcement learning. Authors show that even if all value functions are linear in a given set of features and the exploration data covers all directions, evaluating any policy might require a sample size that is exponentially large in the problem horizon. This is an interesting and somewhat surprising result, and I believe it would be of interest to the wider RL community. I recommend acceptance of this paper.